# Role of Global Self-Esteem in Predicting Life Satisfaction of Nursing Students in Poland, Spain and Slovakia

**DOI:** 10.3390/ijerph17155392

**Published:** 2020-07-27

**Authors:** Ewa Kupcewicz, Elżbieta Grochans, Marzena Mikla, Helena Kadučáková, Marcin Jóźwik

**Affiliations:** 1Department of Nursing, Collegium Medicum, University of Warmia and Mazury in Olsztyn, 10-719 Olsztyn, Poland; 2Department of Nursing, Pomeranian Medical University in Szczecin, 71-210 Szczecin, Poland; grochans@pum.edu.pl; 3Department of Nursing, University of Murcia, Campus de Espinardo, 30100 Murcia, Spain; marmikla@yahoo.com; 4Murcian Institute of Biosanitary Research (IMIB), 30120 Murcia, Spain; 5Department of Nursing, Faculty of Health, Catholic University in Ruzomberok, 034-01 Ruzomberok, Slovakia; helena.kaducakova@ku.sk; 6Department of Gynecology and Obstetrics, Faculty of Medicine, Collegium Medicum, University of Warmia and Mazury in Olsztyn, 10-045 Olsztyn, Poland; marcin.jozwik@uwm.edu.pl

**Keywords:** global self-esteem, life satisfaction, student, nursing

## Abstract

*Background:* This study analyzed the role of global self-esteem and selected sociodemographic variables in predicting life satisfaction of nursing students in Poland, Spain and Slovakia. *Methods*: The study subjects were full-time nursing students from three European countries. A diagnostic survey was used as a research method, while the Rosenberg self-esteem scale (SES) and the satisfaction with life scale (SWLS) were used to collect data. *Results:* The research was performed on a group of 1002 students. The mean age of those surveyed was 21.6 (±3.4). The results showed significant differences both in the level of the global self-esteem index (F = 40.74; *p* < 0.0001) and in the level of general satisfaction with life (F = 12.71; *p* < 0.0001). A comparison of the structure of results demonstrated that there were significantly fewer students with high self-esteem in Spain (11.06%) than in Poland (48.27%) and in Slovakia (42.05%), while more students with a high sense of life satisfaction were recorded in Spain (64.90%) than in Poland (37.87%) or in Slovakia (47.44%). A positive, statistically significant correlation was found between global self-esteem and satisfaction with life in the group of Slovak students (r = 0.37; *p* < 0.0001), Polish students (r = 0.31; *p* < 0.0001) and Spanish students (r = 0.26; *p* < 0.0001). Furthermore, a regression analysis proved that three variables explaining a total of 12% output variation were the predictors of life satisfaction in Polish students. The regression factor was positive (ßeta = 0.31; R^2^ = 0.12), which indicates a positive correlation and the largest share was attributed to global self-esteem (9%). In the group of Spanish students, global self-esteem explained 7% (ßeta = 0.27; R^2^ = 0.07) of the output variation and 14% in the group of Slovak students (ßeta = 0.38; R^2^ = 0.14). *Conclusions:* The global self-esteem demonstrates the predictive power of life satisfaction of nursing students, most clearly marked in the group of Slovak students. The measurement of the variables under consideration may facilitate the planning and implementation of programs aimed at increasing self-esteem among young people and promoting the well-being of nursing students.

## 1. Introduction

The literature explores various approaches to self-esteem. According to Rosenberg, self-esteem is a positive or negative attitude towards oneself, a kind of global self-assessment. At the same time, the author points out that high self-esteem is a belief that one is “good enough”, a valuable person, while low self-esteem means dissatisfaction with oneself, a kind of self-rejection [1]. Self-esteem is a complex and multifaceted concept, often used interchangeably with self-evaluation [2,3]. By its very nature, it is a subjective construct, based on personal perception and assessment. It involves not only the emotional, but also the performative aspect of functioning. It is a representative indicator of health and well-being, as well as a variable explaining human behavior [3,4].

Research shows that people with high self-esteem experience more positive emotions and are very active, persistent and healthier. On the other hand, people with low self-esteem experience more negative emotions and show less activity and even an attitude of avoiding difficulties, challenges and risks [3]. There are theoretical assumptions and empirical evidence that global self-esteem is a feature or state. According to Mark Leary’s theory, self-esteem reflects the sociometric position of a person in a group. In turn, self-esteem as a state reflects the level of social approval and acceptance, as well as the sense of group membership and is susceptible to changes under the influence of mood or effort put into the task [5]. Studies show that self-esteem changes during human life, increases between late childhood and adolescence and then increases during late adolescence and early adulthood [6]. This is a period related to a young person’s education and the formation of new competences with regard to their personal vision of life. It is important that positive feelings, a lack of negative feelings and the level of satisfaction with life should be part of the subjective well-being in the life of every person [7].

The literature offers numerous studies on the evaluation of life satisfaction as a result of comparing one’s own situation against the standards set by a given person. The evaluation of life satisfaction is expressed in the sense of satisfaction with one’s own achievements and conditions [8,9]. As the research shows, the level of self-esteem and satisfaction with life among nursing students (as well as among active nurses) varies and it is usually at a moderate level [10,11,12,13]. Many researchers have indicated that the variables in question reveal important relationships proving, among other things, the level of individual activity [14,15,16,17,18]. Farwa et al. searched for a link between self-esteem and life satisfaction and socioeconomic status in a group of nursing students at the University of Lahore. The results showed a positive correlation between socioeconomic status and self-esteem and student life satisfaction, which, according to the authors of the research, will translate into a more effective study [14]. Other studies conducted among a group of 348 Chinese students (214 men and 134 women) showed a close relationship between self-esteem and life satisfaction for both women and men. Furthermore, the impact of the socioeconomic status on the life satisfaction of the students was observed [15]. An interesting study was conducted at the University of Cyprus to determine the relationship between religiousness, self-esteem, stress and depression among students of nursing, social care and early education. Self-esteem in that study played a significant role because higher levels of self-esteem in students were associated with lower levels of depression, while the strength of religious and spiritual beliefs negatively correlated with depression [16]. The links between self-esteem and student health were also confirmed by Karaca et al. Self-esteem, academic satisfaction, stress and negative events over the past year have been shown to have a strong link to mental health in a group of 516 Turkish nursing students [17]. Other Turkish studies at the Foundation University in Istanbul attempted to determine the impact of four years of nursing studies on the self-esteem and assertiveness of academic youth. It was found that the level of self-esteem in the examined students increased in the fourth year of studies, while the level of assertiveness in the students varied depending on the year of studies, increasing in the second and third year of studies [18].

The relationship between global self-esteem and sociodemographic variables such as age, marital status, level of education and professional experience of nursing students is also evident, as was shown by Shresth et al. [19]. The research results also show the relationship between emotional intelligence and self-esteem. The results of a study carried out on a group of 400 nursing students at Kafrelsheikh University in Egypt can provide an example here. Significant positive correlations between variables were observed and it was shown that emotional intelligence and self-esteem are important factors determining student progress. Most the examined students demonstrated low and moderate self-esteem [20]. Other studies, in turn, show that self-esteem is significantly related to social functioning and plays a significant role in shaping the image of the nursing profession [11]. Based on the literature review, it can be concluded that self-esteem shows predicted relationships with emotional dispositions, predispositions determining readiness to take action, as well as aspects of task-oriented and social functioning [6].

In line with the presented theoretical assumptions, the main objectives of this study were established:To identify differences in the level of global self-esteem and life satisfaction between nursing students from Poland, Spain and Slovakia;To determine the role of global self-esteem and selected sociodemographic variables, i.e., age, year of study and gender in the prediction of life satisfaction of nursing students in Poland, Spain and Slovakia. Through scientific investigations, the following research problems were formulated;Are there any differences in the level of global self-esteem and life satisfaction among nursing students from Poland, Spain and Slovakia and, if so, to what extent?What is the role of global self-esteem and selected sociodemographic variables, i.e., age, year of study and gender in predicting life satisfaction of nursing students in Poland, Spain and Slovakia?

The following research hypotheses were put forward:It is assumed that nursing students have different levels of global self-esteem and life satisfaction depending on their country of origin;There is a relationship between global self-esteem and selected sociodemographic variables, i.e., age, year of study and gender and satisfaction with life in nursing students from different countries.

## 2. Materials and Methods

### 2.1. Settings and Design

Between May 2018 and June 2019, a diagnostic survey was carried out with the participation of 1002 students enrolled in first degree-undergraduate, full-time studies in the nursing program at the University of Warmia and Mazury in Olsztyn, the Pomeranian Medical University in Szczecin (Poland), the University of Murcia in Murcia (Spain) and the Catholic University in Ružomberok (Slovakia). The criteria for inclusion in the study included having the status of a nursing student, age up to 30 and expressed consent to participate in the study. The criteria for exclusion from the study were the period of the examination session and the absence of consent to participate in the study. The research was carried out at the place where the teaching classes for students are held, after obtaining permission from the teacher conducting the classes. One of the researchers delivered the prepared sets of questionnaires to the universities where the research project was carried out. Students were provided with information about the purpose of the study and instructions on how to fill in the answer sheet and they had the opportunity to ask questions and receive explanations. After expressing informed consent to participate in the study, students were given a set of questionnaires. On average, it took about 20 min to complete the questionnaire. The research was anonymous and voluntary and the students could withdraw from the study at any time. A total of 1017 sets of questionnaires were distributed. After collecting the data and eliminating defective questionnaires, 1002 (i.e., 98.5%) correctly completed questionnaires were qualified for further analysis. The collected material was entered in Excel software and the results were analyzed collectively.

### 2.2. Participants

The study involved 1002 students, including 404 (40.3%) from Poland, 208 (20.8%) from Spain and 390 (38.9%) from Slovakia. The mean age of all the respondents was 21.60 years (±3.40). The distribution of women and men in different countries was significantly different (*p* < 0.001). Women accounted for 91.32% (*n* = 915) of all surveyed persons and men only 8.68% (*n* = 87). The distribution of the number of students in particular years of studies in the analyzed groups was similar. The number of first-year students was 329 (32.83%), the second year: 458 (45.71%) and the number of third-year students was 215 (21.46%). The age of the respondents was analyzed in three age groups: ≤20 years, 21–22 years and ≥23 years. The distribution of age groups by country was significantly different (*p* < 0.001). Among Spanish students, 73.08% were 20 and below (Table 1).

### 2.3. Research Instruments

The research applied the diagnostic survey method and two research tools (validated and available for general use in the mother tongue in each of the countries) were used to measure variables:Global self-esteem scale (SES) by *Moriss Rosenberg* [3,4,6];Satisfaction with life scale (SWLS) by Ed Diener, Robert A. Emmons, Randy J. Larsen, Sharon Griffin [9].

A self-constructed questionnaire was used to collect sociodemographic data, such as place of residence (country), gender, age, level of education, form and year of study.

#### 2.3.1. M. Rosenberg’s Global Self-Esteem Scale SES

Rosenberg SES self-esteem scale is made up of 10 statements that relate to beliefs and are diagnostic in their nature. The examined person indicates the extent to which he/she agrees with each of them by providing answers on a four-point scale from 1 to 4, which indicate: 1—strongly agree, 2—agree, 3—disagree, 4—strongly disagree. Following the assumed method of evaluating the answers, statements that are positively formulated are reversed: 1, 2, 4, 6, 7, so that the highest score is awarded for answers expressing a higher level of self-assessment. The result is the sum of points, which is an indicator of the overall self-esteem level. The range of possible results is from 10 points to 40 points. A higher score reflected higher self-esteem. Raw results were converted into standard units on the sten scale. The SES scale has good psychometric properties, with Cronbach’s alpha ranging from 0.81 to 0.83 [3,4,6].

#### 2.3.2. Satisfaction with Life Scale—SWLS

The satisfaction with life scale (SWLS) contains five statements and is used to measure life satisfaction expressed in the sense of satisfaction with one’s achievements. The respondent indicates to what extent each of the statements refers to his/her previous life, providing a response on a seven-point scale from 1 to 7, which indicate: 1—strongly disagree, 2—disagree, 3—slightly disagree, 4—neither agree nor disagree, 5—slightly agree, 6—agree, 7—strongly agree. The result is a sum of points, which is an overall indicator of the sense of satisfaction with life. The range of results is from 5 points to 35 points. A higher score indicates greater satisfaction with life. Raw results were converted into standard units on the sten scale. The SWLS scale has good psychometric properties and a reliability factor (Cronbach’s alpha) is 0.87 [9].

### 2.4. Statistical Analysis

The data generated during the study were subjected to statistical analysis using the Polish version of Statistica 13 (TIBCO, Palo Alto, CA, USA). Socio-demographic data are presented as the number of cases and as the percentage values and the distribution of variables in groups for individual countries was checked with the chi-squared (χ^2^) test. The overall indicator of global self-esteem was converted to standardized units, which were interpreted according to the characteristics of the sten scale. It contains 10 units and the scale jump equals 1 sten. Sten scores between 1 and 4 were considered low, between 5 and 6 were considered average and between 7 high and 10 high [6,9]. Differences in average global self-esteem and life satisfaction results among students by country of origin were tested with the ANOVA (F) test, while intergroup differences were tested with the post hoc test. The r-Pearson correlation coefficient was used to determine the relationship between the variables. Multiple regression analysis was used to build a model of estimation of a random variable from explanatory variables. The interpretation of the strength of the relationship between the variables was based on Guilford’s classification. In all tests, the significance level *p* < 0.05 was assumed [21].

The presented research results are part of a larger international research project [22]. The research meets the criteria for a cross-sectional study [23], and the project received approval (No. 4/2020) from the Senate Committee on Ethics of Scientific Research at the Olsztyn University.

## 3. Results

### 3.1. Diversification of Global Self-Esteem and Life Satisfaction Results in Nursing Students

The results of research on global self-esteem and life satisfaction conducted in Poland, Spain and Slovakia indicate that global self-assessment is related to the subjective well-being of nursing students. Taking into account the cultural conditions in individual countries, significant differences were observed in nursing students both for overall global self-esteem index (F = 40.74; *p* < 0.0001) and for overall life satisfaction (F = 12.71; *p* < 0.0001) (Table 2). Detailed analyses with the post hoc test (NIR test) showed that the level of global self-esteem among Spanish students was significantly lower than among Polish (*p* < 0.0001) and Slovak students (*p* < 0.0001). However, no significant differences in the level of the overall global self-esteem index were found between students from Poland and Slovakia (Table 2; Figure 1 and Figure 2). Subsequent analyses with a post hoc test showed that the general level of life satisfaction among nursing students in Poland was significantly lower than in Slovakia (*p* < 0.03) and lower than in Spain (*p* < 0.0001). In turn, students from Slovakia demonstrated a significantly lower rate of satisfaction with life than students from Spain (*p* < 0.002) (Table 2, Figure 1 and Figure 2). Within a given country, no significant differences in the average results for global self-esteem or sense of satisfaction with life were noted in relation to selected sociodemographic characteristics such as age, gender and year of study.

After converting the raw results into sten-scale standardized units, it was found that the distribution of low, average and high global self-esteem in nursing students varied significantly from country to country (χ^2^ = 103.66; *p* < 0.0001). As shown by the analyses, the number of respondents with low self-esteem was significantly higher in Spain (37.02%) than in Poland (27.97%) and Slovakia (26.15%). On the other hand, students with high self-esteem were significantly less numerous in Spain (11.06%) than in Poland (48.27%) and Slovakia (42.05%) (Figure 3).

Subsequently, raw results obtained for the sense of life satisfaction of nursing students were converted into standard units on the sten scale. As in the case of global self-esteem, there were statistically significant differences in the distribution of low, average and high levels of satisfaction with life in nursing students in Poland, Spain and Slovakia (χ^2^ = 44.21; *p* < 0.0001). Students with a high sense of life satisfaction were significantly more numerous in Spain (64.90%) than in Poland (37.87%) and Slovakia (47.44%) (Figure 4).

### 3.2. Correlations between Global Self-Esteem and Life Satisfaction

Subsequent analyses were associated with the calculation of Pearson’s linear correlation coefficients (r) between the overall global self-esteem index and life satisfaction of nursing students, determining the strength and direction of the relationship. In the group of Slovak students, a statistically significant positive correlation (r = 0.37; *p* < 0.0001) between global self-esteem and satisfaction with life was observed on an average level (Figure 5). The same direction and a similar strength of relationship at the average level were found for Polish students. The correlation coefficient was r = 0.31 (*p* < 0.0001) (Figure 6). The lowest correlation coefficient (r = 0.26; *p* < 0.0001) was observed among Spanish students (Figure 7). These results indicate that nursing students with higher global self-esteem are significantly more satisfied with life, regardless of their country of residence.

### 3.3. Predictors of Life Satisfaction of Nursing Students

Further analyses attempted to determine the predictors of life satisfaction among the examined nursing students in individual countries. When constructing the multiple regression model, life satisfaction was assumed as the explained (dependent) variable and a range of sociodemographic variables, i.e., age, gender, year of study and global self-esteem, were used as explanatory (independent) variables. Regression analysis showed that three variables explaining a total of 12% output variation were the predictors of life satisfaction in Polish students (Table 3). The regression factor was positive (ßeta = 0.31; R^2^ = 0.12), indicating a positive correlation, with the largest share attributed to global self-esteem (9%). The other two variables, year of study and gender, demonstrated a small share in the prediction of life satisfaction among Polish students (3%). In nursing students in Spain and Slovakia, only one variable—global self-esteem—proved to be a predictor of life satisfaction. In the group of Spanish students, global self-esteem explained 7% (ßeta = 0.27; R^2^ = 0.07) of the output variation and 14% in the group of Slovak students (ßeta = 0.38; R^2^ = 0.14). In both cases, the regression index was a positive value, which means that global self-esteem is positively linked to subjective well-being, which is an important element of health.

## 4. Discussion

The authors of this study have attempted to define the role of global self-esteem in the lives of nursing students in Poland, Spain and Slovakia, recognizing that self-esteem is based on self-knowledge, which affects satisfaction with life. There are numerous factors that determine how nursing students perceive themselves. The image they create of themselves and the attitude they have towards themselves have a strong influence on a wide range of personal and social behaviors.

In this study, significant differences were observed in the level of the overall global self-esteem index among nursing students. Students from Spain obtained lower average values (26.03) of global self-esteem than nursing students from Poland and Slovakia (29.69 vs 29.10). Comparing the mean values obtained in own research with the mean results (30.85) obtained for the data collected in 53 countries by other researchers, it can be concluded that, as in most countries, most the examined nursing students obtained results higher than the arithmetic midpoint of the scale [22].

The average results of the second examined variable, i.e., life satisfaction, were distributed slightly differently. The highest average value was obtained by students from Spain (24.04), while lower values were obtained by students from Slovakia (22.40) and Poland (21.46).

When reviewing the results obtained by other researchers, it can be observed that they indicate different levels of global self-esteem and life satisfaction among the groups of respondents, sometimes being quite diverse [10,24,25,26,27,28,29,30]. Velmurugan et al. proved that 65.3% of the surveyed nursing students revealed moderate self-esteem, while 22.9% had low and only 11.9% had high levels of self-esteem [10].

As demonstrated in the analyses of own research results, students with high self-esteem were significantly less numerous in Spain (11.06%) than in Poland (48.27%) or Slovakia (42.05%). In contrast, more than half (64.90%) of the surveyed students from Spain described their sense of life satisfaction as high, which may indicate that in addition to self-esteem, other factors have a significant impact on their well-being. In turn, in a study in Cyprus, average levels of global self-esteem were found for the majority (71.3%) of nursing students [27].

The results of numerous studies indicate that global self-esteem correlates with life satisfaction [28,29,30]. To confirm this thesis, it is worth citing the results of research carried out by Patel et al., who found a significant impact of self-esteem on the satisfaction with life of Indian students [28]. Similarly, in the research presented in this study, the results of linear correlation indicate that the higher global self-esteem among students is correlated with increased satisfaction with life. The presented study also showed that global self-esteem is a predictor of satisfaction with life (from 7% to 14%) of nursing students, and the strongest predictive power was shown in the Slovak group. However, sociodemographic variables, i.e., age, gender and year of studies did not play a significant role in predicting the satisfaction of the studied students.

Several researchers also sought individual determinants of subjective well-being in early adulthood. It was found that students’ self-esteem, physical appearance and positive everyday events were indicators of satisfaction with life among Spanish youth [29]. In contrast, Williams et al. conducted an interesting study to determine the relationship between daily life satisfaction of nursing students and the body mass index (BMI) and the consumption of food and drink. The study group consisted of 215 students, of whom approximately 44.9% were overweight, obese or extremely obese. It was found that the increase in individual satisfaction with everyday life predicted a 36% decrease in the probability of overweight/obesity [30].

Self-esteem is a personal resource related to the well-being of working nurses and it makes sense to develop it in the next stages of education. A study by Perez-Fuentes et al. of a group of 1073 nurses found that global self-esteem correlates strongly with health behavior. The results of these studies have shown that poor sleep quality and the type of food consumed affect the self-esteem of nurses [31]. In other studies, using a group of 1094 nurses, it was found that global self-esteem had a direct and indirect effect on uncontrolled eating [32]. Abnormal eating behavior often signals problems with self-esteem, acceptance of one’s own body and difficulties with adaptation in the group [33]. Research shows that self-esteem has strong links to the dimensions of emotional functioning. Consequently, it is important to promote the well-being of students. An illustration of this is a project aimed at promoting the mental health of first-year nursing students at the University of Minify (Egypt). As a result of the actions taken, there was an increase in self-esteem and a decrease in the level of student anxiety in relation to the initial parameters [34]. Korean researchers, on the other hand, evaluated a short program for nursing students that focused on promoting positive self-esteem and ego development, as these two variables are related to academic achievement and students’ life satisfaction. As indicated by the authors of the study, the results of self-identity and self-esteem increased significantly in the group of participating students, while the results in the control group remained at the same level [35]. In subsequent Korean studies, an attempt was made to identify factors affecting the learning outcomes of nursing students under simulated conditions. It was found that the self-esteem and collective effectiveness of nursing students during team classes in simulated conditions affects their educational effects [36]. There is evidence suggesting that other variables that are expected to be related to the level of self-esteem are key attributes of healthcare workers, such as interpersonal communication, emotional intelligence and empathy [25]. It is worth highlighting that it is impossible to present all directions in which scientific research on self-esteem and satisfaction with life is developing. The authors of the study have only reviewed the most important findings, which may be helpful in interpreting the results of their own research and comparing them with the results of other researchers, as well as outlining the area of future research on the development of global self-esteem and quality of life of academic youth as future employees of the medical services sector.

It is also worth considering the new situation recently caused by the COVID-19 pandemic, at least in view of the factors related to restrictions which have forced quite a radical change in the functioning of young people, especially in social contacts. In further scientific deliberations, it can be assumed that nursing students with low global self-esteem and low life satisfaction may experience negative emotional states of dissatisfaction in their relationships with others, which may lead, among others, to feelings of loneliness.

### Limitations and Implications for Professional Practice

The results of the research presented in this study show that global self-esteem is a predictor of life satisfaction for nursing students, and its measurement can help to plan and implement programs aimed at increasing the sense of self-esteem of young people and promoting the well-being of students.

The authors of the study indicated some limitations, related to the fact that the research did not exclude students experiencing (at that time) family related, financial or emotional problems, not related to studying. Since the presented study is the first on the international scene in selected European countries, such as Poland, Spain and Slovakia, it requires replication with a larger study group, as well as verification of the relationships established here in other situational contexts.

## 5. Conclusions

Significant differences in the level of overall global self-esteem and life satisfaction of nursing students based on the country of residence were found. Students from Spain achieved lower average values of global self-esteem than nursing students from Poland and Slovakia, but they achieved higher values of satisfaction with life than other students. The percentage shares of low, average and high global self-esteem and life satisfaction among nursing students in particular countries were significantly different. There were fewer students with high self-esteem in Spain than in Poland or Slovakia, while more highly satisfied students were reported in Spain than in Poland or Slovakia.

Although global self-esteem demonstrates the predictive power of life satisfaction of nursing students in all of the analyzed countries, it is most clearly marked in the group of Slovak students. In the Polish group, two sociodemographic variables (year of study and gender) slightly influenced the prediction of life satisfaction in nursing students.

## Figures and Tables

**Figure 1 ijerph-17-05392-f001:**
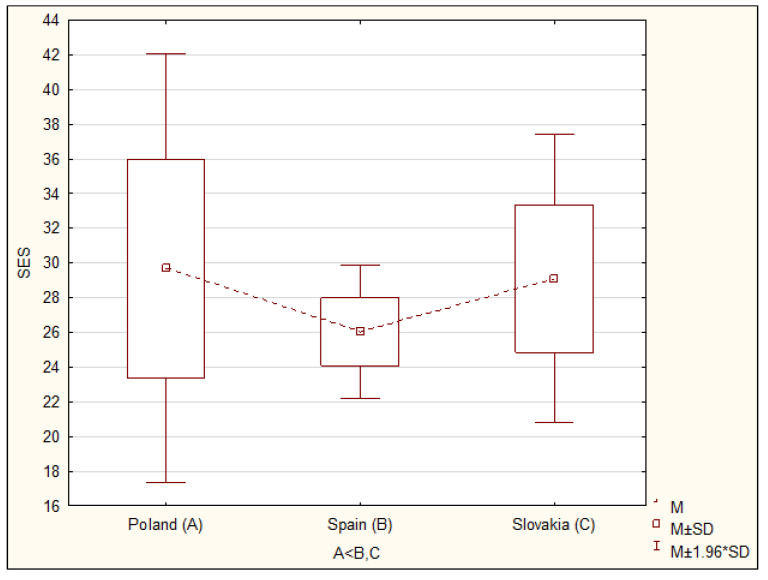
Comparison of average results of life satisfaction according to SES among Polish, Spanish and Slovak students.

**Figure 2 ijerph-17-05392-f002:**
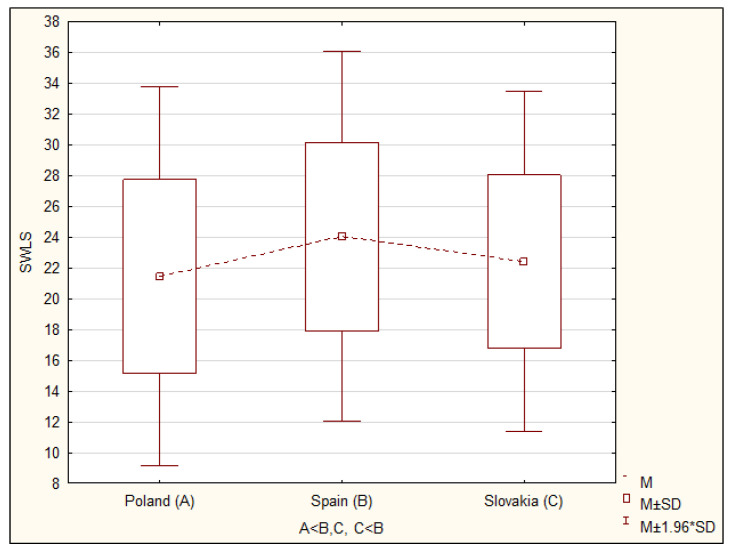
Comparison of average results of life satisfaction according to SWLS among Polish, Spanish and Slovak students.

**Figure 3 ijerph-17-05392-f003:**
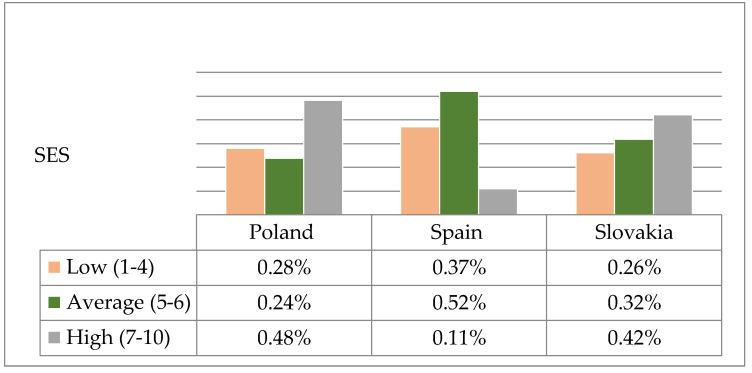
Structure of results of the global self-esteem according to SES on the sten scale among Polish, Spanish and Slovak students.

**Figure 4 ijerph-17-05392-f004:**
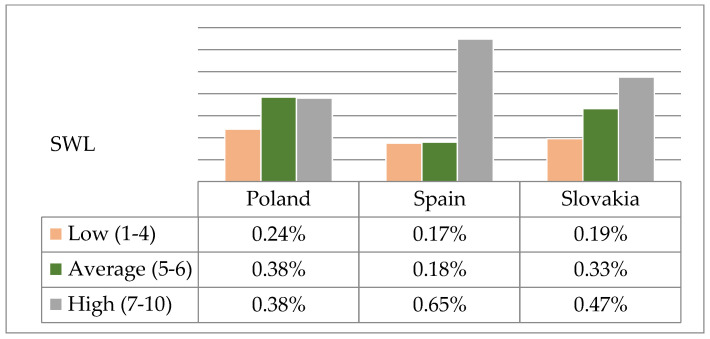
Structure of results of life satisfaction according to SWLS on the sten scale among Polish, Spanish and Slovak students.

**Figure 5 ijerph-17-05392-f005:**
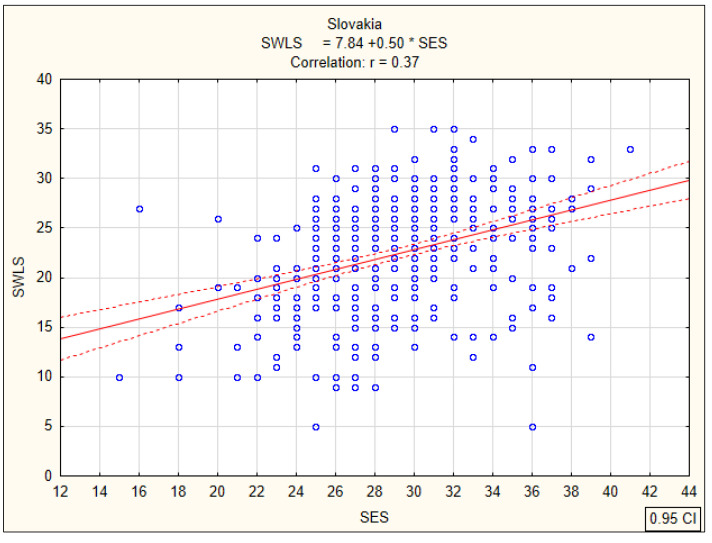
Correlation between global self-esteem according to SES and life satisfaction according to SWLS among Slovak students.

**Figure 6 ijerph-17-05392-f006:**
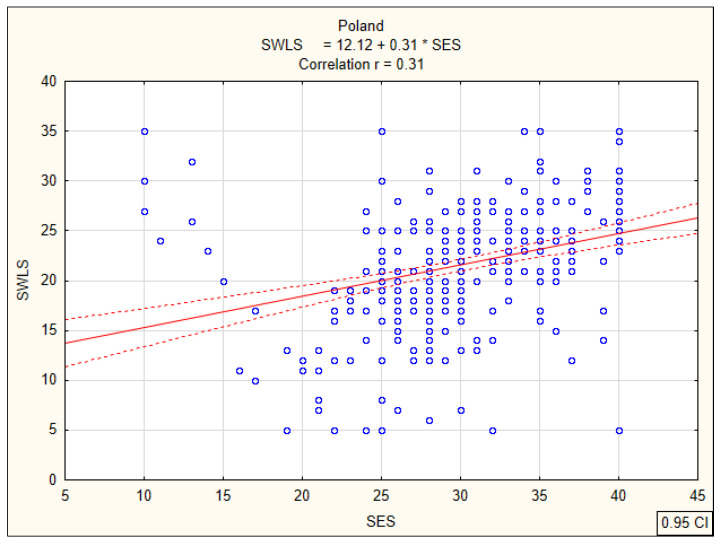
Correlation between global self-esteem according to SES and life satisfaction according to SWLS among Polish students.

**Figure 7 ijerph-17-05392-f007:**
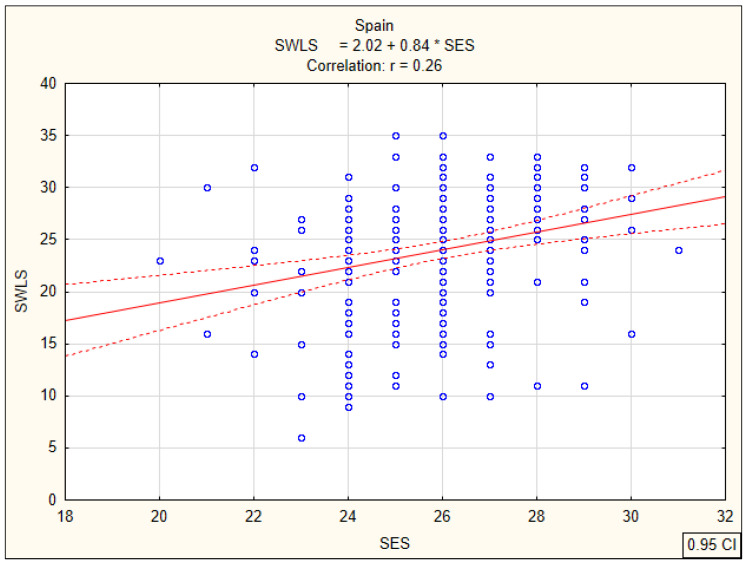
Correlation between global self-esteem according to SES and life satisfaction according to SWLS among Spanish students.

**Table 1 ijerph-17-05392-t001:** Characteristics of the examined group.

Variables		*N* = 1002 (%)	Country of Origin			Chi-Squared Test χ^2^	*p*
			Poland *n* = 404 (%)	Spain *n* = 208 (%)	Slovakia *n* = 390 (%)		
Gender	Female	915 (91.32)	365 (90.35)	171 (82.21)	379 (97.18)	42.72	0.001
	Male	87 (8.68)	39 (9.65)	37 (17.79)	11 (2.82)		
Age (in years)	≤20	401 (40.02)	124 (30.69)	152 (73.08)	125 (32.05)	135.93	0.001
	21–22	410 (40.92)	190 (47.03)	25 (12.02)	195 (50)		
	≥23	191 (19.06)	90 (22.28)	31 (14.9)	70 (17.95)		
Year of studies	first	329 (32.83)	139 (34.41)	64 (30.77)	126 (32.31)	1.16	0.88
	second	458 (45.71)	183 (45.30)	96 (46.15)	179 (45.90)		
	third	215 (21.46)	82 (20.30)	48 (23.08)	85 (21.79)		

Statistically significant: *p* < 0.001.

**Table 2 ijerph-17-05392-t002:** Diversity of the results concerning global self-esteem according to SES and life satisfaction according to SWLS among students in Polish, Spanish and Slovak research.

Variables	Country of Origin			*df*	ANOVA (F-Test)	*p-*Value
	Poland—A *n* = 404 (40.3%)	Spain—B *n* = 208 (20.8%)	Slovakia—C *n* = 390 (38.9%)			
	M ± SD, Me, Min–Max CI ± 95%	M ± SD, Me, Min–Max CI ± 95%	M ± SD, Me, Min–Max CI ± 95%			
SES	29.69 ± 6.30, 30.00, 10.00–40.00, 29.08–30.31	26.03 ± 1.95, 26.00 20.00–31.00, 25.77–26.30	29.10 ± 4.24, 29.00 15.00–40.00, 28.67–29.52	2	40.74	0.0001 B < A,C
SWLS	21.46 ± 6.28, 22.00, 5.00–35.00, 20.84–22.07	24.04 ± 6.13, 25.00, 6.00–35.00, 23.20–24.88	22.40 ± 5.64, 23.00, 5.00–35.00, 21.84–22.96	2	12.71	0.0001 A < B,C C < B

Statistically significant: *p* < 0.001; A, B, C—post hoc analysis (NIR test). Explanation: *N*—number; arithmetic mean; S—standard deviation; Me—median; Min—minimum; Max—maximum; CI ± 95%—confidence interval; SES—global self-esteem; SWLS—life satisfaction, *df*—degrees of freedom.

**Table 3 ijerph-17-05392-t003:** Summary of regression—life satisfaction according to SWLS among nursing students.

Group/Country	Variables	R^2^	ßeta	ß	ß Error	t	*p*-Value
Poland	SES	0.09	0.31	0.3	0.05	6.6	0.0002
	Year of studies	0.01	0.13	1.1	0.40	2.8	0.01
	Gender	0.02	−0.12	−2.5	0.99	−2.5	0.01
	Single			13.1	1.94	6.7	0.00
	R = 0.36; R^2^ = 0.13; corrected R^2^ = 0.12						
Spain	SES	0.07	0.27	0.9	0.21	4.0	0.0007
	Single			4.8	5.73	0.8	0.41
	R = 0.29; R^2^ = 0.09; corrected R^2^ = 0.07						
Slovakia	SES	0.14	0.38	0.50	0.06	8.01	0.0001
	Single			7.84	1.84	4.27	0.0002
	R = 0.38; R^2^ = 0.14; corrected R^2^ = 0.14						

Statistically significant: *p* < 0.01; *p* < 0.001. Explanation: R—correlation coefficient; R^2^—multiple determination coefficient; ßeta—standardized regression coefficient; ß—non-standardized regression coefficient; ß Error—non-standardized regression coefficient error; t—*t*-test value; SES—global self-esteem.

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
