# Peer review of "Role of Global Self-Esteem in Predicting Life Satisfaction of Nursing Students in Poland, Spain and Slovakia"

_ijerph, 2020, doi:10.3390/ijerph17155392_

Round 1
Reviewer 1 Report
Role of global self-esteem in predicting life satisfaction of nursing students in Poland, Spain and Slovakia
(Review)
Main message
The paper looks into the relationship between self-esteem and life satisfaction amongst nursing students in three different countries; Poland, Spain and Slovakia. The study found that there were differing distributions of students with high self-esteem across the three countries. Self-esteem was found to be a significant predictor of life satisfaction in the three countries but to different extents. In addition, gender and year of studies were significant predictors in the Polish group.
General judgment comments
The paper provides a brief background of studies conducted regarding the topic of self-esteem and life satisfaction in nursing students. Examining predictors of life satisfaction and well-being in nursing students is a pertinent area of research in public health, given the crucial role they play as frontline healthcare workers. Understanding and promoting their well-being is even more important in light of the current global pandemic. In addition, a key strength of the study lies in its sample that consists of nursing students from three different countries. As such, the paper can be strengthened by including a deeper analysis of the observed differences in the samples across countries. Overall, the paper requires substantial revisions in order to address these gaps.
Major issues
1) Introduction and literature review
- The introduction provides a rather detailed account of existing studies looking at self-esteem and life satisfaction in nursing. However, a more detailed discussion on the theoretical basis for the relationship between self-esteem and life satisfaction could be made in the introduction before presenting findings in the existing literature.
- In addition, was there a specific rationale for looking at nursing students in particular?
- While the research questions were presented at the end of the introduction, there appears to be no clear hypotheses made with regards to these questions.
- Minor edit: “Are there any differences in the level of global self-esteem and life satisfaction among nursing students from Poland, Spain and Slovakia and, if so, to what extent?” (pg 3)
2) Discussion
- The discussion was largely a summary of findings presented in the Results section and a corroboration with past studies. The present study has significant strength in comparing samples from three different countries but this was not well developed upon in the discussion of the findings. Possible explanations for (i) differences in the predictive power of self-esteem on life satisfaction observed in Slovak students vs the other two groups and (ii) proportion of students with high self-esteem in Spanish group could be proposed for these observations in order to better place the present study in existing work conducted on self-esteem and life satisfaction.
- The link between some of the past studies that were discussed with the present study and the focus on self-esteem and life satisfaction were also not immediately apparent and clear and somewhat tenuous – e.g. self-esteem, health behaviours, food/drink consumption, obesity (pg 10).
- Although authors have pointed out accurately that “It is worth highlighting that it is impossible to present all directions in which scientific research on self-esteem and satisfaction with life is developing”, the implications and future directions of findings from this study could be better elaborated upon in relation to existing measures/studies on the topic. This would help to better highlight the contributions and the place of the current study in the existing body of work on life satisfaction and its predictors in nursing students.
Minor issues
1) Abstract
- First line: “This study analysed the role …” (pg 1)
2) Results
- The posthoc results are described in the text – a table summarizing all the information regarding these posthoc comparisons may be helpful.
- In addition, Table 2 can be accompanied by a figure in order to help readers better visualize the data.
- Degrees of freedom should also be included when reporting the ANOVA and posthoc results.
- “When building the multiple regression model, life satisfaction was assumed to be the explanatory variable, while the pool of explanatory variables was composed of socio-demographic variables, i.e. age, gender, year of study and global self-esteem.” Did you mean that life satisfaction was the dependent/predicted variable?
- Minor edit: Commas were used to represent decimal points in the figures versus periods in text for the decimal points
3) Conclusions
- The conclusion section should be presented in prose
- “Within a given country, socio-demographic variables, such as age, gender and year of study, do not determine the level of global self-esteem and life satisfaction among nursing students.”
However, gender and year of studies were reported to significant predictors in the Polish group in Table 3.
- “The distribution of low, average and high global self-esteem and life satisfaction in nursing students varied considerably in individual countries. There were fewer students with high self-esteem in Spain than in Poland or Slovakia, while more highly satisfied students were reported in Spain as compared to the other countries.”
This statement appears to be contradictory. From the results, it appears that Spain had the lowest proportion of students with low self-esteem and the highest proportion with high self-esteem.
4) Other minor edits
- Cronbach’s alpha (pg 5)
- “In all tests, the significance level p < 05 was assumed” (pg 5)
Final comments
The paper requires a stronger discussion of the theory underlying the study and its results in relation to the countries that the study was conducted in to strengthen its argument.
Author Response
Dear Reviewers,
Thank you very much for a thorough editorial assessment of our manuscript, positive opinions, as well as reviewers’ remarks. We used them as important hints to improve the quality of our paper. Our implemented corrections were done strictly according to the comments. All changes made in the text are marked in yellow. We are enclosing with the re-edited manuscript and cover letter as responses to Reviewers, detailing how we followed their suggestions. Thank you very much for your kind consideration of our paper.
Yours sincerely,
Authors

Reviewer 2 Report
The present paper aims to identify differences in the level of global self-esteem and life satisfaction between nursing students from Poland, Spain and Slovakia. To determine the role of global self-esteem and selected socio-demographic variables, i.e. age, year of study and gender in the prediction of life satisfaction of nursing students in Poland, Spain and Slovakia.
This study employs a robustness analysis. However, the introduction and discussion are not persuasive enough that the findings make a significant contribution to the literature and could, therefore, override these limitations. I include some comments below related to this summary for consideration.
- In relation to the contribution of the study to the literature, I did not get a sense from the article that the findings revealed anything other than what we already know. Please clarified that;
- The introduction of the paper was very descriptive, it did not situate the current study in literature or highlight what the gap in the literature is that this study is trying to address. At least, the authors should situate better the main purposes of this study;
- The discussion is very descriptive and any statements about the contribution and conclusions of the study are not new. At least this moment. Please clarified better and justified your choices.
- Overall, the paper has conditions for be accepted in IJERPH, however the authors should clarified the points above.
Author Response
Dear Reviewers,
Thank you very much for a thorough editorial assessment of our manuscript, positive opinions, as well as reviewers’ remarks. We used them as important hints to improve the quality of our paper. Our implemented corrections were done strictly according to the comments. All changes made in the text are marked in yellow. We are enclosing with the re-edited manuscript and cover letter as responses to Reviewers, detailing how we followed their suggestions. Thank you very much for your kind consideration of our paper.
Comments:
The introduction of the paper was very descriptive, it did not situate the current study in literature or highlight what the gap in the literature is that this study is trying to address. At least, the authors should situate better the main purposes of this study;
Response:
The authors of the paper have made every effort to provide the Introduction in a more informative manner. They have deepened the theoretical assumptions and empirical evidence on global self-esteem and life satisfaction and formulated the research hypotheses.
Comments:
The discussion is very descriptive and any statements about the contribution and conclusions of the study are not new. At least this moment. Please clarified better and justified your choices.
Response:
The authors of the paper have made every effort to provide the Discussion in a more informative manner and have modified their research conclusions.
Since the presented study is the first on the international scene in selected European countries, such as Poland, Spain and Slovakia, it requires replication with a larger study group, as well as verification of the relationships established here in other situational contexts.
Yours sincerely,
Authors
Reviewer 3 Report
This is an interesting study on the prediction of life satisfaction, in nursing studies.
The work is well done and the objectives are clear.
However, due that the study has been conducted in students, why did not the authors considered to address the impact of their findings in the learning results of the students?
Another think that it could be worth to consider is if there were differences in the results depending on the year of studies. There is a process of maturation, that could impact in the measures done in the study.
Author Response
Dear Reviewers,
Thank you very much for a thorough editorial assessment of our manuscript, positive opinions, as well as reviewers’ remarks. We used them as important hints to improve the quality of our paper. Our implemented corrections were done strictly according to the comments. All changes made in the text are marked in yellow. We are enclosing with the re-edited manuscript and cover letter as responses to Reviewers, detailing how we followed their suggestions. Thank you very much for your kind consideration of our paper.
Comments:
This is an interesting study on the prediction of life satisfaction, in nursing studies.
The work is well done and the objectives are clear.
However, due that the study has been conducted in students, why did not the authors considered to address the impact of their findings in the learning results of the students?
Another think that it could be worth to consider is if there were differences in the results depending on the year of studies. There is a process of maturation, that could impact in the measures done in the study.
Response:
The empirical data used in the present study are a part of a larger international research project being conducted in Poland, Spain and Slovakai. The authors have scheduled consequent works to analyse the variables depending on students’ health condition, health bahaviors, and learning results. In planned studies, selected socio-demographic variables such as age, sex and study year will be considered as well. The directions of scientific investigation indicated by the Reviewer will be an important hint to our further considerations.
Thank you very much for your kind consideration of our paper.
Yours sincerely,
Authors

Round 2
Reviewer 1 Report
The authors have appropriately edited the text
Author Response
Dear Reviewers, Thank you very much for the thorough editorial assessment of our manuscript, positive opinions, as well as the reviewers’ remarks. We used them to improve the quality of our paper. We have enclosed the re-edited manuscript and a cover letter with responses to the Reviewers, detailing how we have implemented their suggestions. As suggested by one Reviewer, final linguistic corrections were carried out by a native speaker from a professional translation office (OSCAR TRANSLATIONS in Olsztyn, Poland). We have also attached a proofreading certificate. Thank you very much for your kind consideration of our paper. Yours sincerely, Ewa Kupcewicz, Ph.D.

Reviewer 2 Report
I would like to thank you for the authors since they addressed all of my comments. Therefore, I suggest that the paper could be accepted for publication in its current form.
Author Response

(The authors gave the same response as above.)

Reviewer 3 Report
The authors have not answered the questions that I have done in my review. Please, can you answer them?
Author Response
Dear Reviewers, Thank you very much for the thorough editorial assessment of our manuscript, positive opinions, as well as the reviewers’ remarks. We used them to improve the quality of our paper. We have enclosed the re-edited manuscript and a cover letter with responses to the Reviewers, detailing how we have implemented their suggestions. As suggested by one Reviewer, final linguistic corrections were carried out by a native speaker from a professional translation office (OSCAR TRANSLATIONS in Olsztyn, Poland). We have also attached a proofreading certificate.
Thank you very much for your kind consideration of our paper. Yours sincerely, Ewa Kupcewicz, Ph.D.
The authors would like to explain that in the presented study, within individual countries, no significant differences between levels of global self-esteem and life satisfaction were found depending on the year of study, age and gender of the respondents. In further analyses, it was demonstrated that two variables, i.e. year of study and gender, represented a minimal contribution (3%) to the prediction of life satisfaction only in the group of Polish students. In the presented study, the authors did not analyse the effect of global self-esteem and life satisfaction on the learning results of nursing students because the empirical data used in the paper constitute a part of a more extensive international research project conducted in Poland, Spain and Slovakia and these data will be a subject of interest in the future.